# Improvement in Intestinal-Failure-Associated Liver Disease by Using Parenteral Fish Oil as Monotherapy: Case-Based Review of the Literature

**DOI:** 10.3390/reports6020028

**Published:** 2023-06-12

**Authors:** Smaragdi Fessatou, Afroditi Kourti, Nikolaos Zavras, Sofia Zouganeli, Niki Kouna, Eustathios Stefos, Ino Kanavaki

**Affiliations:** 1Department of Paediatric Gastroenterology, Hepatology and Nutrition, 3rd Department of Paediatrics, Attikon University General Hospital, National and Kapodistrian University of Athens, 12462 Athens, Greece; 2Department of Pediatric Surgery, Attikon University General Hospital, National and Kapodistrian University of Athens, 12462 Athens, Greece; 3Department of Nutrition and Dietetics, Attikon University General Hospital, 12462 Athens, Greece; 4Second Department of Anesthesiology, Attikon University Hospital, National and Kapodistrian University of Athens, 12462 Athens, Greece

**Keywords:** fish oil, intestinal-failure-associated liver disease, intravenous lipid emulsion, parenteral nutrition

## Abstract

Intestinal-failure-associated liver disease (IFALD) is a common complication of prolonged parenteral nutrition (PN). Risk factors for IFALD include clinical features, as well as medical interventions, and its management was initially based on the decrease or interruption of parenteral nutrition while increasing enteral nutrition. However, the tolerance of full enteral nutrition in children with intestinal failure may require prolonged intestinal rehabilitation over a period of years. As a consequence, infants unable to wean from PN are prone to develop end-stage liver disease. We describe the case of an infant receiving long-term PN who was diagnosed with IFALD wherein we were able to reverse IFALD by switching lipid emulsions to fish oil monotherapy. A systemic review of case reports and case series on reversing IFALD using fish oil lipid emulsion follows the case description.

## 1. Introduction

Intestinal failure (IF) in infants and children, caused by insufficient bowel length or function, is a devastating condition that can be broadly defined as the inability of the gastrointestinal tract to sustain life without supplemental parenteral nutrition (PN). Parenteral nutrition is a lifesaving therapy for these children, but long-term PN treatment is limited by serious complications, including blood-stream infections, mechanical catheter-associated complications (breakage or thrombosis), metabolic bone disease, metabolic abnormalities, and others. The hepatobiliary consequences of PN include cholestasis, liver inflammation, and fibrosis, which leads to cirrhosis, portal hypertension, end-stage liver disease, and death [1,2].

Up to 75% of infants who require PN for 60 days or more develop intestinal-failure-associated liver disease (IFALD) [1,2]. Up to 26% of these patients will end up with a liver transplant and 27% will eventually die [2].

The risk factors for IFALD include clinical features such as premature birth, low birth weight, gastrointestinal mucosal disease, and recurrent bacterial sepsis, as well as medical interventions, such as long-term use of PN, absence or delayed introduction of enteral feeds, prolonged diverting enterostomies, and multiple operative procedures [2,3,4].

In recent years, it was demonstrated that the composition and timing of enteral feeding can affect the achievement of enteral autonomy. Prompt initiation of enteral feeding after bowel resection has been reported to improve the rate of enteral autonomy. For infants with short-bowel syndrome, human milk is considered most suitable for enteral feeding, because it contains growth factors, amino acids, immunoglobulins, and other immunologically important compounds that may promote intestinal adaptation. When human milk is unavailable, amino acid-based formulas are commonly used [3,4].

The management of children with IFALD was initially based on the decrease or the interruption of parenteral nutrition while increasing enteral nutrition. However, tolerance of full enteral nutrition in children with intestinal failure may require prolonged intestinal rehabilitation over a period of years [3]. As a consequence, infants unable to wean from PN are prone to develop end-stage liver disease [4].

Intravenous lipid emulsions (IVLEs) are indispensable components of PN as a non-carbohydrate source of energy. Soybean-oil-based lipid emulsions (LE) have been widely used for several decades and are still used as the major components in the current lipid formulations. For example, Intralipid^®^ 20% contains 100% soybean oil and the more recent SMOFlipid 20% is a mixture of 30% soybean oil, 30% medium-chain TGs, 25% olive oil, and 15% fish oil. Studies have demonstrated a clear association between IVLEs and PN-associated liver disease. The mechanism for this remains unclear; however, both animal and human research mostly implicates phytosterols and ω-6 (n6) fatty acids [5,6,7,8,9].

In contrast, fish-oil-based lipid emulsions (FOLEs) have been shown to be associated with full resolution of IFALD [10,11,12,13,14,15]. The beneficial effects of FOLE have been attributed to the high proportion of ω-3 fatty acids, which, in contrast to ω-6 fatty acids, have been shown to possess considerable anti-inflammatory properties [16,17,18,19].

We describe the case of an infant receiving long-term PN who was diagnosed with IFALD wherein we were able to reverse IFALD by switching from SMOF to fish oil monotherapy. A systemic review of case reports and case series of reverse IFALD using FOLEs follows the case description.

## 2. Case Report

A girl born prematurely, at 29 weeks of gestation with a birth weight of 1300 g, was diagnosed on day 9 with necrotizing enterocolitis. A laparotomy was performed and extensive bowel necrosis was found, involving nearly the whole small bowel. Massive small bowel resection was performed with only the duodenum and 15 cm of the small intestine spared. Her general condition improved but she still could not tolerate oral feeding. Cholestasis with progressively increasing bilirubin levels was noted at 3 weeks after initiation of TPN.

The patient was transferred to our unit at the age of 5 months. Oral feeding was started and was gradually increased, until the patient could cover 30% of her total calory needs, with respective decrease in the amount of PN injected. However, cholestasis showed no improvement and the patient begun showing signs of biliary cirrhosis, hypersplenism, and coagulopathy. A liver biopsy could not be performed due to coagulopathy so liver ultrasound and elastography were used to determine the severity of liver damage. At that time, the patient had a liver stiffness result equal to 8 kPa (F2 fibrosis—moderate). Table 1 summarizes the laboratory results from the day she was admitted to our hospital, Figure 1 and Figure 2 demonstrate the bilirubin and transaminase levels’ evolution, and Figure 3 shows the patient’s growth chart.

At the age of 15 months, the patient’s course was complicated because of catheter-related sepsis and multiorgan failure. Bilirubin rose up to 19, 94 mg/dl, liver elastography was 10 kPa (F3 fibrosis—severe), and liver transplantation was discussed. SMOF infusion was decreased with no improvement. The replacement of LE by FOLE did not happen until the child was 19 months old (due to difficulties to getting access to them), with a starting dose of 0.5 g/kg/d for 2 weeks, followed by 1 g/kg/d.

At the age of 21 months jaundice was improved and TB/DB levels finally normalized at 24 months of age. FOLE monotherapy was maintained for 8 months. SMOF was then reintroduced and maintained as a combination of 30% FOLE and 70% SMOF lipids.

The patient is now 3.5 years old and still dependent on PN. Liver function is fully preserved and we have not seen any relapse of cholestasis.

## 3. Search Strategy

We searched PubMed, PubMed Central, Scopus, Web of Science, and ScienceDirect up to 31 December 2021, combining the keywords intestinal-failure-associated liver disease; parenteral-nutrition-associated liver disease; parenteral nutrition lipid formulations; and fish-oil-based lipid emulsions. We further searched the reference lists of identified articles for additional papers. We included case reports or case series of pediatric patients. We restricted results to English language published papers and this could be considered as a limitation of the study.

## 4. Discussion

Over the last few decades, the approach for children with liver disease due to intestinal failure has evolved and this fact led to a decrease in cases of end-stage liver disease requiring liver transplantation. One of the most important things that led to this change is the different approach to the use of lipids in parenteral nutrition, either by reducing the amount used or by using alternative sources of lipids, with fish oil lipid emulsions proving to be the most important development in parenteral nutrition.

The mechanisms by which lipid emulsions participate in IFALD development are not fully understood. There is strong evidence that the long-chain polyunsaturated fatty acids (LCPUFAs) within the soy-based lipid emulsions play a pivotal role in the pathogenesis of IFALD [5,6,7,8,9]. The most important LCPUFAs are ω-6FA and ω-3FA, that have common metabolic pathways. They interact with each other through a negative feedback pathway by competing for nutrient substrate availability and for the same metabolic enzymes for membrane synthesis and integration. A crucial factor in the reduction in inflammatory response is the ω-6FA to ω-3FA ratio (n6:n3 ratio). It is known that active ω-3FAs interfere with the metabolization of the ω-6 fatty Acids (FA) arachidonic acid, leading to a downregulation of inflammatory eicosanoids [6]. In order to act as an immunomodulator the optimal ratio of n6:n3 is thought to be between 1:1 and 4:1 [9].

Over the last decades, soy-based lipid emulsions (Intralipid) have been the cornerstone of PN, and contain predominantly ω-6FAs with a n6:n3 ratio of 5.5:1. Intralipid contains ω-3FAs in the form of a-linolenic acid, and not in the biologically active form of docosahexanoic and eicosapentaenoic acid, and thus does not provide a substantial and utilizable source of ω-3FAs to the infant, because infants have a limited capacity to metabolize a-linolenic acid. Furthermore, the predominant ω-6FAs have been implicated in the development of hepatic steatosis, which is one of the early hallmarks of IFALD.

PN solutions, which are composed of ω-3FAs in addition to ω-6FAs (SMOFlipid 20%), have several beneficial effects concerning the prevention and treatment of IFALD. Additionally, it has recently been shown that the use of fish-oil-based lipid emulsions could reverse hepatic steatosis in both PN and non-PN models of hepatic steatosis. The administration of ω-3FA through several mechanisms, such as a reduction in phytosterols dose, by eicosanoid-mediated mechanisms and by modifying biliary canalicular membrane, could affect bile flow. Moreover, the addition of ω-3FA, through the reduction in ω-6FA, leads to a change in the profile of eicosanoids from pro-inflammatory to anti-inflammatory. This shift in inflammatory mediators may have an important role in the progression of hepatitis and the resultant fibrosis in response to the initial cholestatic and steatotic insult [5,6,7,8,9].

FOLE monotherapy was first described by Gura et al. in 2002 [12], in a PN-dependent, soy-allergic adolescent who developed severe essential fatty acid deficiency after a period of fat-free PN administration. Since 2004, fish oil IVLE monotherapy has been prescribed to treat patients with IFALD.

We report the reversal of IFALD observed in a child receiving long-term PN by replacing SMOF with FOLE. This reversal was seen during the period of FOLE treatment, while the patient remained PN-dependent and despite the fact that she had factors predisposing to the failure of FOLE therapy.

Several clinical studies have previously described the reversal of cholestasis in PN-dependent infants with IFALD after changing lipid emulsion to FOLE. In most reports, infants developed IFALD due to soybean oil (Intralipid) administration [10,11,20,21,22,23,24,25,26,27,28,29,30,31,32,33,34]. In only two cases, as in our case, IFALD developed while receiving SMOF as a lipid source [35,36] (Table 2).

In the case we describe, the infant progressed to severe liver disease despite low-dose IVLE (SMOF 1 gr/kg/d) and PN cycling. Switching LE to FOLE was decided upon as a rescue therapy and a bridge to liver transplantation. Based on previous publications and the published literature, fish-based IVLE was initiated at 0.5 g/kg/d and was gradually increased to a maximum of 1 g/kg/d [10,11,15].

Previously published case reports mention that the median time for resolution of IFALD varies from 40 days to 8 months [11,23,24,25,26,27,29,31,34]. More specifically, one of the first cases reported was that of an infant who received fish-oil-based IVLE after developing cholestasis and IFALD. The authors documented the resolution of IFALD after 8 months of therapy [25]. FOLE was also used in a retrospective cohort of 12 children diagnosed with short bowel syndrome (SBS) and advanced IFALD. In that cohort, FOLE was associated with liver function restoration within a median of 24 (range 15.3–55.3) weeks [26]. In another paper describing two infants who received FOLE, the authors reported that, regardless of the enteral nutrition regimen, there was a complete resolution of hyperbilirubinemia and cholestasis within 8 weeks of the initiation of FOLE [11]. Our patient had a complete resolution after 20 weeks of fish-oil-based IVFE therapy, without any side effect. Liver transplantation is therefore no longer a plan.

More importantly, our patient had poor prognostic factors, such as low birth weight, advanced age at fish-oil initiation, severe liver disease, and other comorbidities, such as renal disease. FOLE initiation reversed IFALD in 5 months and the child remains asymptomatic after one year of single SMOF-LE use.

Nandivada P et al. reviewed the data of patients treated with FOLE at Boston Children’s Hospital, Boston, MA, USA, from 2004 to 2014, and whose cholestasis failed to reverse despite fish oil ILE monotherapy, in order to identify the factors associated with the failure of fish oil therapy in treating IFALD and to guide referral guidelines [38]. The authors noted that, among the 182 patients treated with fish oil ILE, 86% achieved cholestasis resolution and 14% failed therapy. The patients who failed therapy had a lower birth weight and were older at FOLE initiation (20.4 weeks [9.9, 38.6 weeks]) compared to those whose cholestasis resolved (11.7 weeks [7.3, 21.4 weeks]). Moreover, the patients who failed therapy had more advanced liver disease at FOLE initiation, as evidenced by a higher direct bilirubin (10.4 mg/dL [7.5, 14.1 mg/dL] vs. 4.4 mg/dL [3.1, 6.6 mg/dL]).

We decided to proceed with a combination of SMOF-FOLE infusion in order to benefit from the effects of fish oil while reducing the risk of essential fatty lipid deprivation. Furthermore, a published review of the long-term outcomes of patients with IFALD that biochemically resolved under FOLE, and who resumed exclusive soybean oil LE, concludes that they were at high risk of IFALD relapse [34].

## 5. Conclusions

Without question, the most effective treatment for IFALD is to maximize enteral nutrition while decreasing PN. However, a significant number of patients are unable to tolerate enteral feeding at the moment needed. Fish-oil-based lipid emulsion has promise in treating IFALD, but there are limited data regarding its use in children, especially as monotherapy. There is a need for further randomized control trials to formulate a standardized protocol for the administration of such emulsions.

## Figures and Tables

**Figure 1 reports-06-00028-f001:**
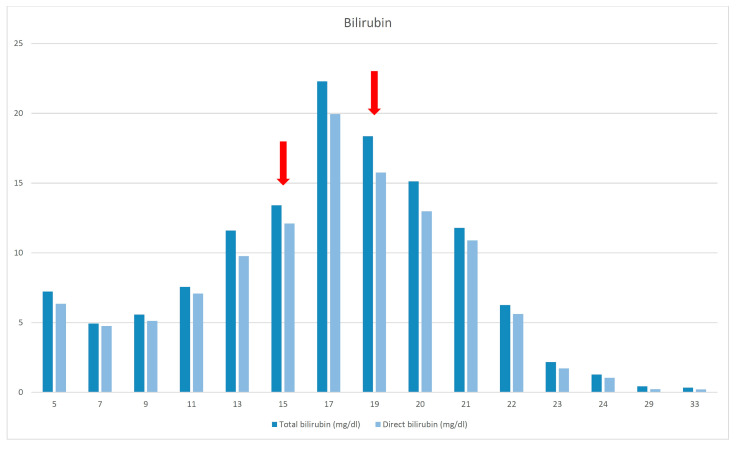
Bilirubin levels. The red arrows (1st) indicates the sepsis episode and the introduction of FOLE (2nd).

**Figure 2 reports-06-00028-f002:**
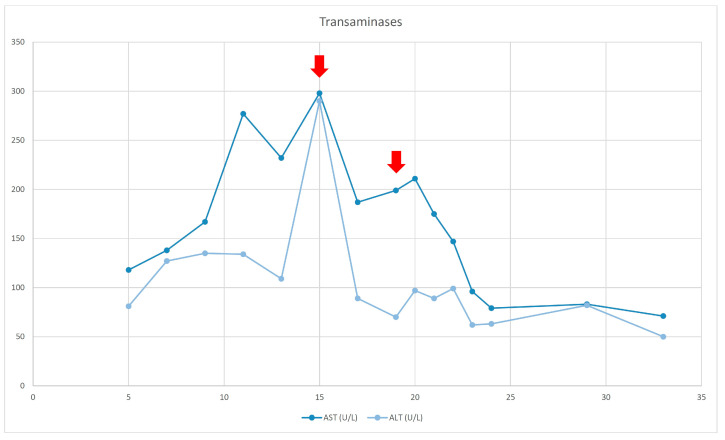
Transaminases evolution. The red arrows (1st) indicates the sepsis episode and the introduction of FOLE (2nd).

**Figure 3 reports-06-00028-f003:**
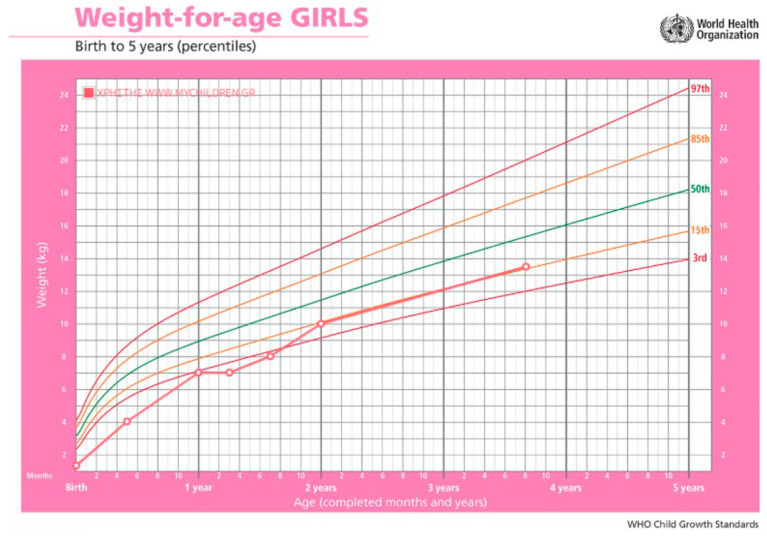
Patient’s growth chart. The green line represents the 50th percentile for weight according to age. The orange lines represent the 15th percentile and the red lines the 97th percentile.

**Table 1 reports-06-00028-t001:** Serum liver function tests and coagulation studies.

Age (Months)	Total Bilirubin (mg/dL)	Direct Bilirubin (mg/dL)	AST (Aspartate Aminotransferase) (U/L)	ALT (Alanine Aminotransferase) (U/L)	Platelets (K/UL)	INR (International Normalized Ratio)
5	7.23	6.35	118	81	165,000	1.06
7	4.94	4.74	138	127	170,000	1.13
9	5.57	5.12	167	135	130,000	1.11
11	7.56	7.08	277	134	116,000	1.23
13	11.6	9.77	232	109	102,000	1.16
15 (catheter related sepsis)	13.4	12.1	298	290	107,000	0.99
17	22.3	19.94	187	89	94,000	1.21
**19 (initiation of ** **fish oil lipid** **emulsion)**	**18.36**	**15.76**	**199**	**70**	**70,000**	**1.20**
20	15.12	12.98	211	97	110,000	1.26
21	11.79	10.89	175	89	96,000	1.15
22	6.26	5.62	147	99	120,000	1.23
23	2.17	1.72	96	62	131,000	1.16
24	1.27	1.04	79	63	140,000	1.07
29	0.44	0.23	83	82	153,000	1.09
33	0.33	0.20	71	50	170,000	1.08

**Table 2 reports-06-00028-t002:** Summary of case reports and case series identified by systemic literature review, with the case reports of patients that were on SMOF before the initiation of FOLE in bold.

No.	First Author	Year/Ref	Conclusions	No. of Patients
1.	Gura KM et al. DOI:10.1542/peds.2005-2662	2006/[11]	This new therapy may offer a potential solution in the treatment or prevention of hepatotoxicity in parenteral-nutrition-dependent patients and may provide an alternative therapy to avoid morbidity, mortality, and the need for liver/small bowel transplantation in children and adults who are dependent on PN and provide the time necessary for bowel adaptation.	2
2.	Gura KM et al. DOI:10.1542/peds.2007-2248	2008/[25]	Parenteral fish-oil-based fat emulsions are safe and may be effective in the treatment of parenteral-nutrition-associated liver disease.	18
3.	Ekema G et al. DOI:10.1016/j.jpedsurg.2008.01.005	2008/[23]	In the light of accumulating evidence, it is hoped that, in the future, Omega-3 fatty acids will gradually gain credit not only in the cure of PNAC but also as a novel protective strategy of the liver during TPN.	1
4.	Puder M et al. DOI:10.1097/SLA.0b013e3181b36657	2009/[15]	Fish-oil-based ILE is safe, may be effective in treating PNALD, and may reduce mortality and organ transplantation rates in children with SBS.	42
5.	Diamond IR et al. DOI:10.1097/MPG.0b013e318182c8f6	2009/[24]	Omegaven is associated with the restoration of liver function in patients with SBS and advanced liver disease. Parenteral omega-3 fatty acids, such as Omegaven, have the potential to fundamentally alter the paradigm of neonatal SBS, from one of early death or transplantation from liver failure to a more chronic disease. More children with SBS should achieve full enteral tolerance and those who do not have the capacity for intestinal adaptation should be able to survive and receive an intestinal graft when they are older.	12
6.	de Meijer VE et al. DOI: 10.1177/0148607109332773	2009/[37]	Our clinical studies showed that a parenteral fish-oil-based lipid emulsion, as monotherapy in a subset of pediatric patients dependent on PN, is safe and efficacious in reversing PNALD and normalizing EFAD status.	90
7.	Lee SI et al. DOI:10.1203/PDR.0b013e3181bbdf2b	2009/[20]	The findings may indicate an added benefit of reduced triglyceride levels for patients treated with fish oil and this effect coincides with markers for improved liver function and nutritional status.	18
8.	Rollins MD et al. DOI: 10.1177/0884533610361477	2010/[26]	Temporary elimination of SLE and supplementation with enteral fish oil improved cholestasis in PN-dependent infants. Further trials are needed to evaluate this management strategy.	6
9.	Chung PHY et al. DOI: 10.1055/s-0029-1238283	2010/[27]	Omega-3 fatty acid offers new possibilities for the management of patients with short bowel syndrome. Bowel adaptation can have a successful outcome if cholestasis can be averted. Clinicians should be alerted about this treatment protocol before considering an invasive surgical bowel-lengthening procedure or transplantation.	4
10.	Le HD et al. DOI: 10.1177/0148607110371806	2010/[13]	A fish-oil-based lipid emulsion used as monotherapy in children who exclusively depended on PN for survival was associated with significant improvement in all major lipid panels as well as improvement in hyperbilirubinemia. Parenteral fish oil may be the preferred lipid source in children with dyslipidemia.	10
11.	Sant’Anna AMGA et al. DOI: 10.1155/2012/571829	2012/[28]	The authors report a positive experience with the implementation of a multidisciplinary approach and with the use of FOE in infants with SBS and severe PNALD. The earlier the FOE was initiated during the cholestatic process, the shorter the time to resolution.	4
12.	Premkumar MH et al. DOI: 10.1016/j.jpeds.2012.10.019	2013/[14]	Younger gestational age infants demonstrated higher degrees of cholestasis, longer time to resolution of cholestasis, and increased mortality. Higher levels of cholestasis were associated with longer time to resolution. Fish oil lipid emulsion monotherapy led to the resolution of cholestasis in all surviving infants.	57
13.	Premkumar MH et al. DOI: 10.3945/an.113.004671	2013/[29]	Our experience with the use of FOLE in PNALD continues to be encouraging. Prematurity continues to be a major determinant in mortality and the severity of cholestasis. This calls for further controlled studies designed to optimize the dose and timing of intervention in the use of FOLE in neonates.	97
14.	Calkins KL et al. DOI: 10.1177/0148607113495416	2013/[21]	A limited duration of fish oil appears to be safe and effective in reversing intestinal-failure-associated liver disease.	10
15.	Nandivada P et al. DOI: 10.1097/SLA.0000000000000445	2015/[22]	Cirrhosis from PNALD may be stable, rather than progressive, once cholestasis resolves with FO therapy. Furthermore, these patients may not require transplantation and show no clinical evidence of liver disease progression, even when persistently PN-dependent.	51
16.	Strang BJ et al. DOI: 10.1177/0884533616643697	2016/[30]	Fish-oil-based IVFE was effectively used to reverse PNAC in a child with SBS. Despite early STEP, the patient was not able to tolerate enteral feedings and required bowel tapering. This case illustrates that early surgical intervention did not allow for improved feed tolerance. This resulted in a significant period without enteral nutrition, leading to the development of cholestasis. The use of fish-oil-based IVFE may permit a longer duration of PN administration without the development of cholestasis or liver disease, allowing for longer time for bowel adaptation prior to the need for surgical intervention.	1
17.	Nandivada P et al. DOI: 10.3945/ajcn.116.137083	2016/[38]	Most infants with IFALD responded to FO therapy with resolution of cholestasis, and liver transplantation was rarely required. Early FO initiation, once biochemical cholestasis is detected in parenteral nutrition-dependent patients, is recommended.	182
**18.**	**Lee S et al.** **DOI: 10.1177/0148607114567200**	**2016/[35]**	**In conclusion, 2 infants with advanced IFALD showed reversal of cholestasis by switching from SMOF to Omegaven monotherapy.**	**2**
19.	Belza C et al. DOI: 10.1016/j.jpedsurg.2017.01.048	2017/[31]	The use of Omegaven^®^ is associated with reduced cholestasis and inflammation, but with persistence or worsening of fibrosis in some patients. A subset of patients with progressive fibrosis may develop portal hypertension and progressive liver disease.	6/40
20.	Sorrell M et al. DOI:10.1097/MPG.0000000000001397	2017/[32]	The use of intravenous fish LEs in premature infants appears to be safe and reverses PNALD, despite significant liver disease and intestinal failure. This therapy should be used in preterm infants with PNALD and followed long term to evaluate development.	13
21.	Zhang T et al. DOI: 10.1038/s41430-018-0096-z	2018/[33]	Fish oil therapy alleviates IFALD in children.	32
22.	Wang C et al. DOI: 10.1002/jpen.1463	2019/[34]	In this study, FO effectively treated cholestasis, and SO resumption was associated with cholestasis redevelopment in nearly a quarter of subjects. Long-term FO may be warranted to prevent end-stage liver disease.	48
**23.**	**Lee S et al.** **DOI: 10.3390/jcm9113393**	**2020/[36]**	**The reversal of IFALD in preterm infants on combination lipid emulsion containing fish oil was achieved by switching to fish oil monotherapy.**	**15**
24.	Gura KM et al. DOI: 10.1016/j.jpeds.2020.09.068	2021/[39]	FOLE recipients experienced a higher rate of cholestasis resolution, lower aspartate aminotransferase to platelet ratio index, and fewer liver transplants compared with SOLE recipients. This study demonstrates that FOLE may be the preferred parenteral lipid emulsion in children with intestinal-failure-associated liver disease when DB reaches 2 mg/dL.	189

## Data Availability

The clinical data of this case report are available in the patient’s medical record. Due to personal data protection rules, data cannot be published.

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
