# Peer review of "Improvement in Intestinal-Failure-Associated Liver Disease by Using Parenteral Fish Oil as Monotherapy: Case-Based Review of the Literature"

_reports, 2023, doi:10.3390/reports6020028_

Round 1
Reviewer 1 Report
1.The title is misleading. “Reversal of intestinal failure–associated liver disease by using parenteral fish oil as monotherapy” suggests causality (= fish oil reverses liver disease). However, this is a massive over-interpretation of the findings. The authors may state “Improved liver enzymes in association with fish oil intake in an infant with intestinal failure–associated liver disease”.
2.The abstract is horrible. You need to define all acronyms that you are using (e.g. PN, SMOF…), and the findings need to be more clearly described (e.g., key findings from the case).
3.You need to be more cautious in interpreting the findings from the case. You cannot state that you reverse IFALD by fish oil, you simply show improvement of selected biomarkers, and fish oil was one of the measures taken. Liver biopsy results and/or non-invasive tests (e.g. LiMAX, transient elastography) would be useful.
4.The case is complex and confounded by episodes of sepsis, which makes it particularly challenging to assess the effect of fish oil. Table 1 and the figures (by the way, there are figure legends missing!) need to indicate when fish oil therapy was initiated and which clinical events (e.g. catheter-associated sepsis, antibiotic therapy etc) relate to the liver enzymes.
5.The weight and growth of the infant should be documented as well, alongside the daily calorie intake.
No major issues, but acronyms need to be introduced when used. The lab values in the table and figures need proper referencing ("7.7" instead of "7,7").
Author Response
Please see the attachement

Reviewer 2 Report
Authors described in a case report with systematic review of literature, a case of a preterm infant receiving long-term parenteral nutrition with diagnosis of IFALD with a initial switching from SMOF to fish oil monotherapy.
It is a nice case report, well described and discussed.
I have some comments during the review process before the publication:
1) some references are missing:
- lines 28-31
- lines 37-41 (in addition I suggest to improve these concepts. Preterm birth is a cause of total parenteral nutrition administration, due the immaturity of intestinal tract. It could be nice to describe the side effects, in long and brief term, of TPN. In addition, I suggest to describe the recently demonstrated positive effects on metabolic conditions of minimal enteral feeding in case of parenteral nutrition administration)
-lines 51-52
-lines 124-126
2) Figures are not introduced in the text. In addition, please modify and add the unit of measurement in the abscissa and order not in the legend.
3) In table 1 and 2, why some lines are in bold characters? I suggest in table 2 to describe the type of population (preterm? Young patients?). The title of the article is not necessary, maybe could be better add “type of study” and “conclusions”. It’s not easy to summarize conclusion in this systematic review.
4) typing errors in lines 112 and 129
5) limitations of these systematic review should be added (by example the selection of only English study)
6) please revise and correct the abbreviations in the text
Author Response
Please see the attachement

Round 2
Reviewer 1 Report
no further comments
Author Response
Thank you for your review.
Reviewer 2 Report
Congratulations
Author Response
Thank you for your review!